# Imaging Diagnostics and Pathology in SARS-CoV-2-Related Diseases

**DOI:** 10.3390/ijms21186960

**Published:** 2020-09-22

**Authors:** Manuel Scimeca, Nicoletta Urbano, Rita Bonfiglio, Manuela Montanaro, Elena Bonanno, Orazio Schillaci, Alessandro Mauriello

**Affiliations:** 1Department of Biomedicine and Prevention, University of Rome “Tor Vergata”, Via Montpellier 1, 00133 Rome, Italy; manuel.scimeca@uniroma2.it; 2San Raffaele University, Via di Val Cannuta 247, 00166 Rome, Italy; 3Saint Camillus International University of Health Sciences, Via di Sant’Alessandro, 8, 00131 Rome, Italy; 4Nuclear Medicine Unit, Department of Oncohaematology, Policlinico “Tor Vergata”, viale oxford 81, 00133 Rome, Italy; n.urbano@virgilio.it; 5Department of Experimental Medicine, University of Rome “Tor Vergata”, Via Montpellier 1, 00133 Rome, Italy; bonfiglio.rita@gmail.com (R.B.); manuela.montanaro@uniroma2.it (M.M.); elena.bonanno@uniroma2.it (E.B.); alessandro.mauriello@uniroma2.it (A.M.); 6Fondazione Umberto Veronesi (FUV), Piazza Velasca 5, 20122 Milano, Italy; 7Diagnostica Medica’ & ‘Villa dei Platani’, Neuromed Group, 83100 Avellino, Italy; 8IRCCS Neuromed, Via Atinense, 18, 8607 Pozzilli, Italy; 9Tor Vergata Oncoscience Research (TOR), University of Rome “Tor Vergata”, 00133 Rome, Italy

**Keywords:** SARS-CoV-2, pandemic, imaging diagnostic, pathology, artificial intelligence

## Abstract

In December 2019, physicians reported numerous patients showing pneumonia of unknown origin in the Chinese region of Wuhan. Following the spreading of the infection over the world, The World Health Organization (WHO) on 11 March 2020 declared the novel severe acute respiratory syndrome coronavirus-2 (SARS-CoV-2) outbreak a global pandemic. The scientific community is exerting an extraordinary effort to elucidate all aspects related to SARS-CoV-2, such as the structure, ultrastructure, invasion mechanisms, replication mechanisms, or drugs for treatment, mainly through in vitro studies. Thus, the clinical in vivo data can provide a test bench for new discoveries in the field of SARS-CoV-2, finding new solutions to fight the current pandemic. During this dramatic situation, the normal scientific protocols for the development of new diagnostic procedures or drugs are frequently not completely applied in order to speed up these processes. In this context, interdisciplinarity is fundamental. Specifically, a great contribution can be provided by the association and interpretation of data derived from medical disciplines based on the study of images, such as radiology, nuclear medicine, and pathology. Therefore, here, we highlighted the most recent histopathological and imaging data concerning the SARS-CoV-2 infection in lung and other human organs such as the kidney, heart, and vascular system. In addition, we evaluated the possible matches among data of radiology, nuclear medicine, and pathology departments in order to support the intense scientific work to address the SARS-CoV-2 pandemic. In this regard, the development of artificial intelligence algorithms that are capable of correlating these clinical data with the new scientific discoveries concerning SARS-CoV-2 might be the keystone to get out of the pandemic.

## 1. SARS-CoV-2 Pandemic

In December 2019, physicians reported numerous patients showing pneumonia of unknown origin in the Chinese region of Wuhan [1]. Thanks to genomic investigations of the pathogen related to these diseases, Chinese health authorities demonstrated that the pneumonia outbreak was correlated to the infection of a new coronavirus, whose genetic sequence is homologous to that of the coronavirus causing severe acute respiratory syndrome coronavirus-2 (SARS-CoV-2) [2]. Following the spreading of the infection in all the world, The World Health Organization (WHO) on 11 March 2020 declared the novel SARS-CoV-2 outbreak a global pandemic.

Despite the SARS-CoV-2 infection appearing very complex, the initial common clinical manifestation of SARS-CoV-2-related disease, which facilitated patient’s detection, was pneumonia [1,2]. Several studies described the molecular mechanisms involved in the infection of pulmonary epithelium by SARS-CoV-2 as well as the immune-mediated response (Figure 1) [2], but still, little is known about infections in non-pulmonary sites. While the latest literature provides insight into clinical manifestations of SARS-CoV-2 disease, histopathology and autopsy findings currently remain scarce. Similarly, imaging diagnostic analysis, such as computerized tomography (CT), have well described pulmonary abnormalities, but still not the involvement of other organs.

SARS-CoV-2 infections are clinically characterized by two phases correlated with a different immune response [3]. During the incubation stages, no severe clinical manifestations are generally observed in healthy population; these subjects are characterized by an appropriate genetic background for specific adaptive immune response, which frequently proves to be competent to eliminate the virus, precluding disease progression to severe stages [3]. However, if the immune response in positive patients does not eliminate the virus, subjects go through the most severe stages of disease, which are characterized by a damaging inflammatory response mainly involving lungs and resulting in severe diffuse alveolar damage (DAD). At this stage of disease, some other organs with high ACE2 expression may be involved. The worse outcome of SARS-CoV-2 includes older age (e.g., >65 years), concomitant cardiovascular disease, hypertension, diabetes, obesity, kidney disease, cancer, and other immunodeficiency conditions [3].

Indeed, recent clinical reports also describe gastrointestinal symptoms, heart injury, vasculitis, kidney dysfunction, and thrombocytopenia in SARS-CoV-2 positive patients [4,5].

A better understanding of the histological changes observed during SARS-CoV-2 infection may increase our knowledge of the pathogenesis of the disease. The identification of specific histological characteristics of the SARS-CoV-2-related diseases, including biomarkers expression, along with specific morphological and molecular alterations detected by imaging methods, will help to formulate an earlier diagnosis and therefore to establish the most appropriate therapeutic protocols in order to prevent the frequent complications caused by this virus.

In this scenario, the combination of imaging diagnostic data and pathological features of SARS-CoV-2-related disease can lay the foundation for developing new diagnostic and therapeutic approaches for the management of SARS-CoV-2 patients.

## 2. From Pathology to Imaging Diagnostic

Currently, the diagnosis of SARS-CoV-2 related diseases is performed by real-time polymerase chain reaction (RT-PCR) analysis on virus RNA and by imaging diagnostic, mainly by chest CT [6,7]. However, other diagnostic procedures can be useful especially for identifying non-lung lesions such as vascular, liver, and kidney alterations. It is of great interest the possibility to identify and understand the ability of SARS-CoV-2 to induce systemic alteration following its infection. Indeed, it is now clear that patients with SARS-CoV-2 frequently died from multi-organ failure and the impairment of the cardiovascular system [8]. In this scenario, the retrospective re-evaluation of imaging diagnostic procedures in asymptomatic patients can provide essential knowledge for a better understanding of the biology of SARS-CoV-2 and its pathogenesis. In fact, the identification of the tissues and organs affected by SARS-CoV-2, as well the study of the molecular characteristics of lesions by molecular imaging analyses, could clarify some aspects of SARS-CoV-2 infection.

On the other hand, the scientific community is exerting an extraordinary effort to elucidate all aspects related to SARS-CoV-2, such as the structure, ultrastructure, invasion mechanisms, replication mechanisms, or drugs, mainly through in vitro studies. Thus, the clinical in vivo data can provide a test bench for new discoveries in the field of SARS-CoV-2, finding new solutions to fight the current pandemic. During this dramatic situation, the normal scientific protocols for the development of new diagnostic procedures or drugs are frequently not completely applied in order to speed up these processes.

In this context, interdisciplinarity is fundamental. Specifically, a great contribution can be provided by the association and interpretation of data derived from medical disciplines based on the study of images, such as radiology, nuclear medicine, and pathology [9,10]. Additionally, the histopathological characterization of tissues of SARS-CoV-2 patients, by both biopsy and autopsy, has made it possible to elucidate some mechanisms related to the SARS-CoV-2 infection. The first image of the SARS-CoV-2 that showed the structure of the virus in the world was caught using a transmission electron microscope [1]. Both the histological and ultrastructural analysis of lung, kidney, heart, and vascular compartments are elucidating the tissue alteration induced by SARS-CoV-2 infection. Furthermore, the use of ancillary techniques, such as immunohistochemical or in situ hybridization analyses, can provide molecular information related not only to the presence of the virus, but also to the virus-related cellular adaptations or, even more importantly, the inflammatory infiltrate associated to SARS-CoV-2 infection. These cellular and molecular biomarkers may also constitute a substrate for developing: (a) new diagnostic protocols based on radiotracers for PET or SPECT investigations, (b) predictive and prognostic assays, (c) new drugs, and/or (d) re-evaluation of already approved drugs for others diseases or viral infections.

Therefore, here, we highlighted the most recent histopathological and imaging data concerning the SARS-CoV-2 infection in lungs and others human organs such as the kidney, heart, and vascular system. In addition, we evaluated the possible match among data of radiology, nuclear medicine, and pathology departments in order to support the intense scientific work against the SARS-CoV-2 pandemic. In this regard, the development of artificial intelligence algorithms that are capable of correlating these clinical data with the new scientific discoveries concerning the SARS-CoV-2 might be the keystone to get out of the pandemic.

## 3. Histological Characteristics of SARS-CoV-2 Infection

The autopsy remains the gold standard to determine the histological lesions associated to SARS-CoV-2 death. Recently, several papers have been published on histological changes observed in subjects who died from SARS-CoV-2, although the number of analyzed cases is very small, and sometimes, they are just case reports.

Overall, the autopsy findings support the concept that the pathogenesis of severe SARS-CoV-2 disease involves the direct viral-induced injury of multiple organs, in particular lungs and heart, and it is often associated to a diffuse coagulopathy [11,12,13,14]. In addition to acute pathologic changes attributed to SARS-CoV-2 virus, chronic changes need to be considered as well, which predispose to a fatal course virus-related disease and through a nonspecific secondary change related to hypossiemia or sepsis.

## 4. Lung Pathology

Lung pathology has been studied in almost all autoptic cases of patients who died from SARS-CoV-2, as reported in the Table 1. In all cases, lungs were macroscopically heavy and blueish-red in color with a diffuse consolidation of the parenchyma. At the histological examination, the hallmark of SARS-CoV-2 involvement was the presence of a DAD characterized by the following: patchy, mild interstitial thickening by edema, extensive intra-alveolar fibrin deposits with the formation of hyaline membranes, marked hyperplasia and desquamation of alveolar epithelium, and the accumulation of macrophages with frequent multinuclear giant cells in association with a severe capillary congestion and a variable inflammatory infiltrate. Moreover, in the late stage, the proliferation of fibroblasts and early collagen fiber deposits within the intra-alveolar exudate was found.

In some cases, a superimposed bronchopneumonia was observed as result of bacterial superinfection and not as a direct result of SARS-CoV-2-induced lung tissue damage.

A focal vasculitis and capillaritis associated to microthrombi were frequently detected in alveolar capillaries, which are associated to DAD. Some authors believe that a neutrophilic, exudative capillaritis of small interstitial vessels with microthrombosis and a relatively small parenchymal inflammation represent the early pulmonary damages before the appearance of an evident DAD [15].

Ackermann et al. [16] analyzed pulmonary autopsy specimens from seven patients who died from SARS-CoV-2 infection and those of seven patients who died from pneumonia caused by influenza A—virus subtype H1N1. In addition to the DAD with necrosis of alveolar lining cells, in lungs from both groups of patients, pneumocyte type 2 hyperplasia, intra-alveolar fibrin deposition, and fibrin thrombi in the alveolar capillaries have been observed, even though the number of capillary microthrombi were nine times more prevalent in SARS-CoV-2 positive patients, as compared to patients with influenza A. Electron microscopy investigations showed endothelial cells with a disruption of intercellular junctions, cell swelling, and a loss of contact with the basal membrane; in this scenario, is possible to hypothesize a fundamental role of endothelial lesions in the pathogenesis of endothelialitis and thrombosis in the lungs of patients with SARS-CoV-2 infection. The presence of SARS-CoV-2 virus within the endothelial cells suggests that direct viral effects may contribute to the endothelial injury [16,17]. The endothelial damage is probably mediated by the angiotensin-converting enzyme 2 (ACE2), which is an integral membrane protein that appears to be the host-cell receptor for SARSCoV-2 [18], with broad mRNA expression in human tissues and high levels of protein detectable on alveolar epithelial cells, intestinal epithelium, and endothelial cells. To support the hypothesis of ACE2 involvement, a greater number of ACE2-positive cells in lungs of patients with SARS-CoV-2 infection were found, as compared to control subjects without SARS-CoV-2 infection.

An interstitial thrombotic necrotizing capillary injury syndrome characterized by the endothelial cell necrosis of capillaries and intraluminal fibrin deposition were described by Magro et al. [19], who examined the lungs and cutaneous tissues of two patients with SARS-CoV-2 infection and severe respiratory failure through autopsy analysis. In contrast to other reports of SARS-COV-2 lung pathology, the authors did not observe the diffuse alveolar damage, hyaline membranes, and pneumocyte involvement that are hallmarks of typical ARDS. The lung injury was restricted only to septal capillaries. Contemporary immunohistochemical investigations demonstrated a deposition of C5b-9, as well as C4d and MASP2, within the microvasculature of the interalveolar septa, which is consistent with activation of the alternative pathway (AP) and lectin pathway (LP) of complement and suggesting that at least in a subset of patients with SARS-COV-2 infection, a complement-mediated thrombotic microvascular injury syndrome occurs.

In addition to microthrombosis of the alveolar vessel, a thrombosis of large and medium-size pulmonary arteries was frequently found. Lax et al. [20] observed a pulmonary arterial thrombosis in 11 SARS-COV-2 positive patients with fatal outcome, despite them receiving prophylactic anticoagulant therapy, while eight cases reported an association with a pulmonary infarction. A central pulmonary thrombosis and embolism deriving from the deep veins of the lower extremities has been reported also in four of the 12 cases studied by Wichmann et al. [21]. In addition, Ackermann et al. [16] showed the presence of thrombi in pulmonary arteries with a diameter of 1 to 2 mm, without complete luminal obstruction, in four of the seven lungs examined from patients with SARS-CoV-2 infection. From the literature, it is not always possible to distinguish a thrombosis of the large pulmonary arteries from a consequence of pulmonary embolism associated to a deep venous thrombosis, since this one has not been sufficiently investigated. Pathogenetic mechanisms of large pulmonary arterial thrombosis in patients who died from SARS-COV-2 are not yet completely understood. It is possible that, similarly to what was hypothesized in the microthrombosis of small vessels, the thrombi of large vessels are most likely secondary to an endothelial damage related to direct viral infection of the endothelial cells, as an extension of endothelial damage of smaller pulmonary vessels to larger vessels. Moreover, thrombosis of pulmonary large vessels could be also related to SARS-COV-2-associated coagulopathy; indeed, from a clinical point of view, many patients reported elevated D-dimer levels with features of both disseminated intravascular coagulation and thrombotic microangiopathy [22], resulting in widespread microvascular thrombosis that may involve other organs, such as the heart, liver, and skin.

In general, vascular changes are nonspecific and have also been described in other viral infections, such as respiratory syncytial virus, human parainfluenza virus 1, and influenza [23]. Moreover, as reported by Konopka et al. [24], it is unknown how or if hyperoxemic mechanical ventilation and other medical interventions may account for some of the histologic findings. However, the contributions of pulmonary thrombosis, embolism, or their combination, which lead to deaths of SARS-COV-2 patients, is yet not well understood because of the limited number of available autopsy studies.

## 5. Heart Pathology

To date, it is known that advanced age, male sex, and the presence of heart diseases are predictors of higher mortality in SARS-COV-2 infection. An observational study of 416 patients with new coronavirus infection in Wuhan, China seems to suggest that cardiac injury is a recurrent condition in hospitalized patients, and that they are associated with a higher risk of mortality [25]. Among causes of death in a Wuhan cohort, myocardial damage and heart failure contributed to 40% of deaths, either exclusively or in conjunction with respiratory failure [25]. Myocardial injury and mortality have been associated with an increase in troponin levels, which is a specific marker of myocardial damage. In positive COVID subjects without cardiovascular disease and without an increase in troponin values in the blood, the risk of death is less than 5–10%; conversely, the risk of death increases several times in people with cardiovascular disease and increased troponin levels in the blood: 37.5% in those with the presence of elevated troponin levels only and 69.4% in those with both elevated troponin levels and a history of cardiovascular disease. Heart disease can affect one patient out of five, at least according to troponin values. Notably, elevated troponin levels carried a strong prognostic value even in the absence of cardiovascular disease history [25,26]. Heart problems often begin first and independently of the extent of respiratory impairment (pneumonia and respiratory failure). There is probably a “two-way relationship” between the presence of heart disease and the risk of an unfavorable outcome from SARS-COV-2 infection. Previous heart disease has been shown to affect the prognosis of positive SARS-COV-2 subjects, but this could only be one side of the coin. Indeed, in the literature, there are continuous reports of acute cardiac lesions, arrhythmias, and hypotension in infected individuals, especially in those requiring intensive care. Cardiovascular complications are more frequent in patients with more severe forms of infection, which is probably due to a more intense inflammatory response. The pathogenesis of cardiac injury is not well established and probably involves different mechanisms: (a) direct myocardial infection by SARS-CoV-2, mediated by ACE2 receptors that are expressed in the cardiomyocytes and in the vessel endothelium, (b) hypoxemia due to respiratory failure, and (c) inflammatory response correlated to the severe systemic inflammation status. It is possible to hypothesize that the acute vasculitis of the intramyocardial vessels may occur because, in some of the autopsies performed on positive SARS-COV-2 subjects, the presence of microvascular lesions, such as vasculitis, were detected.

The diagnosis of acute cardiac injury and myocarditis was frequently based only on troponin evaluation, without additional clinical evidence. Some autopsy reports have described fulminant myocarditis associated to inflammatory mononuclear infiltrate in myocardial tissue, even though there were no evidence of SARS-COV-2 virus in the myocardium [26,27].

In a recent case report, Yan et al. [28] reported an incongruence between the clinical cardiac abnormalities, which are compatible with a fulminant viral myocarditis, and the cardiovascular pathology findings at autopsy, in which no histological changes of viral myocarditis were detected on histopathologic evaluation.

The heart and the vessels are potential targets for SARS-COV-2; however, at present, there are no findings providing evidence of the direct infection and replication of SARS-CoV2 in the heart cells. In a single case report of a 69-year-old patient with influenza-like symptoms quickly worsening to respiratory distress and cardiogenic shock, the endomyocardial biopsy at electron microscopy showed viral particles in macrophages, but not in cardiomyocytes or other specific cardiac cell types [29]. In addition, Craver et al. [30] reported a case of fatal eosinophilic myocarditis in a healthy 17-year-old male, with no interstitial pneumonia and diffuse alveolar damage, in which post-mortem nasopharyngeal swabs detected SARS-COV-2 virus. No other cause for eosinophilic myocarditis was elucidated.

Further pathological studies and autopsy series will be very helpful to clarify the potential of SARS-CoV-2 to directly infect the myocardium and cause myocarditis

## 6. Renal Pathology

Renal dysfunction is common in SARS-COV-2 infections, even though only from 3 to 7% of hospitalized positive SARS-COV-2 patients in Wuhan proceeded to more severe renal disease [10,31].

The most frequent histological changes were similar to those observed in acute tubular injury and involved mainly the proximal tubules; in particular, swelling of the tubular epithelium to necrosis and regenerating changes with flattened tubular epithelium occurred. Focally, the tubules were filled with proteinaceous masses [30]. The cause of kidney injury in SARS-COV-2 is unclear. In an autopsy study of a single patient with SARS-COV-2 infection and acute oliguric renal failure, through electron microscopy analysis, Farkash et al. [32] identified intracellular viral arrays within proximal tubular epithelial cells, which is consistent with direct infection of the kidney by SARS-CoV-2. The receptor for SARSCoV-2 cellular entry is ACE2, which is present at high concentrations in the brush borders of renal tubular epithelial cells.

## 7. Skin Pathology

Among different targets of the SARS-COV-2 virus, skin can be another candidate tissue for this infection, causing both urticarial rashes and papulovesicular exanthems [33,34,35]. Only few reports have described histological findings of these lesions. Gianotti et al. [33,34] reported the histological analysis of eight cases of skin dermatoses in patients affected by SARS-COV-2 infection in the northern part of Italy. The histology showed a wide spectrum of histopathological lesions. In the exanthematic phase, only a mild spongiosis associated to the presence of dilated blood vessels with a few extravasated red blood cells was observed. In the papular phase, the papillary dermis appears markedly edematous, with abnormally swollen, dilatated capillaries and prominent blood extravasation. The presence of perivascular infiltrates was constant, consisting mainly of cytotoxic CD8 lymphocytes and eosinophils. In two of these patients, nests of intraepidermal Langerhans cells associated with vasculitis and a diffuse coagulopathy in the cutaneous vascular plexus were observed.

A similar histological finding was described in a case report by Zengarini et al. [35]. In addition, in this case, the histological examination did not show any specific changes different from other rashes of viral etiology, except for the presence of extremely dilated vessels in the dermis, which could represent a histological diagnostic finding.

Furthermore, Llamas-Velasco et al. [36] reported livedoid purple lesions along with acrocyanosis in a positive SARS-CoV-2 patient: at the biopsy, an obstructive cutaneous vasculopathy and sweat gland necrosis characterized by the presence of dilated blood vessels filled with hyaline thrombi and a mild neutrophilic component in the papillary dermis were detected; nonetheless, the microbiological cultures and PCR for SARS-CoV-2 infection for the skin biopsy were negative.

Differently, in children, a possible association between SARS-COV-2 infection and Kawasaki disease has been hypothesized [37,38]; similarly to what was observed by Gianotti et al. [23], Kawasaki disease exhibits a characteristic perivascular infiltrate that is mainly composed of cytotoxic CD8 lymphocytes associated to hypereosinophilia [39]. Hypereosinophilia could play an important role in vascular thrombosis, as it has shown this role on the activation of the coagulation cascade [40].

## 8. Central Nervous System

Despite SARS-CoV-2 infection having been associated with many neurological symptoms, few studies have been published. Solomon et al. [41] reported neuropathological findings obtained from autopsies of 18 patients with SARS-CoV-2 infection. The principal neurological symptoms reported by patients were myalgia, headache, and loss of taste. The microscopic examinations showed acute hypoxic damage in the brain and cerebellum of all patients, with loss of neurons in the cerebral cortex, hippocampus, and Purkinje cell layer, but thrombi or vasculitis were not detected. Rare foci of perivascular lymphocytes were found in two specimens, and leptomeningeal inflammation was found in one case. No microscopic abnormalities in the bulbs or olfactory tracts were identified. Although sections of brain tissue were positive for the virus in the molecular test, immunohistochemical analysis did not reveal the presence of the virus in the tissue. The authors of the study hypothesize that the positivity to the molecular test of some brain samples may be attributable to virions present in situ; however, a contamination from viral RNA present in the blood cannot be excluded.

## 9. Other Organs

During autopsy examinations of patients who died from COVID19 infection, some authors detected lesions in the liver, adrenals, and testis, reporting nonspecific histological alterations.

Lazx et al. [20] reported changes observed in the liver during the autopsy of 11 patients. A mild increase in sinusoidal lymphocytic infiltration, sinusoidal dilatation, and steatosis are the pathologic changes frequently reported. In particular, a macrovesicular steatosis was found in all patients, involving 5% to 60% of the hepatocytes. A massive confluent and panlobular hepatocyte necrosis was only observed in one case, while it was focal in another two cases. Causes and mechanisms have not been elucidated but are likely multifactorial, including direct viral attack, hepatotoxicity of therapeutic drugs, hyperinflammatory reactions, pre-existing chronic liver disease, and hypoxemic status.

Iuga et al. [42] recently described in five post-mortem examinations of patients with SARS-COV-2 infection an acute fibrinoid necrosis of arterioles in adrenal parenchyma, adrenal capsule, and in the immediately adjacent periadrenal adipose tissue, without significant inflammation, adrenal parenchymal infarcts, or thrombi. However, the same authors affirm that it was not possible to establish if the adrenal vasculopathy was due to the direct viral cytopathic effect, an immune-mediated injury, or to the hypertensive status of patients.

Yang et al. [43] performed a post-mortem examination of the testes from 12 COVID19 patients using light and electron microscopy. Testes from SARS-COV-2 patients exhibited significant seminiferous tubular injury, reduced Leydig cells, and mild lymphocytic inflammation. Spermatogenesis was not altered.

Table 2 summarizes the main important histological lesions observed during the autopsy of patients who died from SARS-CoV-2 infection.

Interpretation of histological data collected from autoptic investigations, although limited still, could provide the scientific rationale for a better understanding of the clinical features of the SARS-COV-2 infection as well as the identification of biomarkers suitable for in vivo imaging analysis, especially computed tomography (CT) and molecular imaging.

## 10. Radiological Aspects of SARS-CoV-2-Related Diseases

Imaging diagnostics, especially radiology, plays a crucial role in the management of patients affected by SARS-CoV-2 infection. Indeed, CT imaging is currently considered the most appropriate in vivo investigation for the detection of lung abnormalities related to the early stage of SARS-CoV-2 pulmonary infection [44,45,46,47,48,49,50,51]. In addition, several studies showed that serial chest CT imaging at different time intervals can be a powerful toll for effectively assessing the disease progression: from the first diagnosis of SARS-CoV-2 infection until patient discharge [52].

Recent investigations demonstrated familial clusters of pneumonia linked to SARS-CoV-2, which indicated the human transmission of the disease [1,3]. In these clusters, some subjects showed ground glass lung opacification by CT, but no clinical symptoms. Following the RT-PCR analysis, patients were confirmed to be affected by SARS-CoV-2. These discoveries indicate, for the first time, that in some subjects, the SARS-CoV-2 infection shows no clinical signs, demonstrating the existence of asymptomatic patients [53]. Thus, in order to contain the spread of the virus, it is very important that all subjects with a clear history of exposure to the virus, regardless of clinical symptoms, or with some peculiar clinical symptoms, should undergo chest CT or SARS-CoV-2 RT-PCR analysis. In a study of Xu et al., the authors demonstrated the fundamental role of chest CT examination in the initial diagnosis of SARS-CoV2 pneumonia [7]. In fact, they showed that some peculiar CT imaging features, such as clusters of patchy ground glass opacities in bilateral multiple lobular with periphery distribution, can be considered distinctive signs of the SARS-CoV-2 pneumonia in asymptomatic patients also [7].

However, currently, the identification of patients affected by SARS-COV-2 is passive and is prevalently based on symptoms. Unfortunately, this approach is not useful neither for the early identification of symptomatic patients nor for the recognition of asymptomatic ones; in this condition, prevention and control of the epidemic become even more difficult. Despite the overcrowding of COVID hospitals, it would be advisable for asymptomatic patients with a history of SARS-CoV-2 exposure to have access to CT investigation to verify the presence of lung lesions. In particular, high-resolution technique (HRCT) can provide a great support in the early diagnosis of patients with a suspicion of SARS-CoV-2 pneumonia [54]. Indeed, HRCT is considered the most accurate imaging analysis for identifying pathognomic characteristics of interstitial pneumonia, such as ground glass areas, crazy paving, nodules and consolidations, mono- or bilateral, patchy or multifocal, central and/or peripheral distribution, declivous or non-declivous [54]. Other imaging/pathological features, such as pleural or pericardial effusion and mediastinal lymphadenopathy, are also possible.

Nevertheless, analyzing the most recent radiological data about SARS-CoV-2 pneumonia, it is clear that imaging features may be variable and patient-specific. Specifically, about 75% of subjects with bilateral lung [11,12,31,32,33,34,35,36,37,38,39,40,41,42,43,44,45,46,47,48,49,50,51,52,53,54,55] and multilobe involvement were also common [56]. In a case series of 21 patients affected by SARS-CoV-2, ground-glass opacity was the main imaging feature detected by chest CT [57], and 29% of these displayed consolidation [56]. About one out of three of SARS-CoV-2 patients reported a peripheral distribution of ground-glass opacity. Conversely, CT images of the chest showed no valuable pleural effusion, nodules, cavitation, and/or lymphadenopathy [57]. Another investigation reporting 51 SARS-CoV-2 patients showed similar findings [58]: most CT images showed pure ground-glass opacity (77%), followed by ground-glass opacity with reticular and/or interlobular septal thickening (75%), and ground-glass opacity with consolidation (59%) and pure consolidation (55%). Of the 51 cases, 86% showed bilateral lung involvement, and the above findings were peripherally distributed in 86% of cases [58].

A clinical study by Tao Ai and colleagues reported that the sensitivity of chest CT imaging for SARS-CoV-2 was 97% in 1014 cases in which infection was confirmed by RT-PCR [56]. According to this evidence, in a solid case series of 59 SARS-CoV-2 patients from China, the positive rate of initial CT examination was 85.7% [53]. In this study published on The Lancet, Huang et al. reported that the sensitivity of chest CT was higher than leukopenia, lymphocytopenia, and C-reactive protein at the initial stage of onset. Interestingly, the authors observed that in young children and infants, the symptoms were frequently mild and atypical, as compared to adults, but they were detectable by chest CT anyway.

Several clinical data indicate that X-ray can be used in the management of SARS-CoV-2 patients [56]. Despite the sensitivity of chest X-ray being considered too low for the detection of pulmonary involvement in early-stage disease SARS-CoV-2 infection [44,59,60,61], it is evident that, in the health emergency setting, this imaging methodology can be a useful diagnostic tool for “day after day” monitoring of lung abnormalities related to SARS-CoV-2 infection, at least in patients already admitted to intensive care units.

Table 3 summarizes the main characteristics of studies that reported CT data of patients affected by SARS-CoV-2 infection.

Evaluation of the severity of SARS-CoV-2 lung infections by imaging, both CT and X-ray analyses, is very important in order to quickly choose the most appropriate clinical approach, as well as respiratory support for infected patients. Currently, several CT scoring systems and only one X-ray scoring system have been developed to quantify the presence/progression of pulmonary lesions in SARS-CoV-2 patients [45,62]. The X-ray scoring system consists of a simple five-point grading tool that was designed for non-radiologist clinicians [63]. In a paper of Borghesi and Maroldi, the X-ray scoring system was used to quantify the “day after day” lung lesions on patients affected by SARS-CoV-2, demonstrating that it may be useful also in the staging of SARS-CoV-2 infection [60]. Specifically, the authors showed that the proposed X-ray scoring system is able to provide relevant information for clinicians as well as highlighting the role of radiologists in this long battle against the SARS-CoV-2 pandemic.

Recently, lung ultrasound (US), besides CT and X-ray, has emerged as diagnostic tool that is suitable for the detection of lung lesions by SARS-CoV-2 [63]. Surprisingly, a study suggested that US analysis is superior to standard CT for the evaluation of pneumonia or respiratory distress syndrome [64]. In particular, Peng et al. [64] summarized five main clinical findings, including thickening of the irregular pleural line, based on data of lung US on 20 patients with pulmonary infection. In addition, the authors reported a strong association between US imaging features and the disease stages, suggesting its possible use in the monitoring of SARS-CoV-2 infection and progression. In line with this, the Chinese Critical Ultrasound Study Group published critical ultrasound-based recommendations on severe SARS-CoV-2, in which US of the lung features was described in detail [64].

However, currently, CT findings are the main factors for both the diagnosis and prognosis of SARS-CoV-2 infection (Table 4).

Artificial Intelligence (AI) software has been recently developed to facilitate imaging diagnosis, especially for CT and/or X-ray analyses [65,66,67,68]. Advantages in the use of AI software for the evaluation of lung infections are automated measurements of wall thickness for intuitive airway analysis, lobe segmentation, and visualization. In addition to clinical procedures and treatments, AI software, currently considered as a new paradigm for health care, may provide different tools that are built upon machine learning algorithms for supporting the decision-making processes. Thus, AI software is used to classify the diseases according to clinical/imaging characteristics and/or to predict its evolution [69]. In this pandemic, several groups investigated the possibility to use AI for the early diagnosis of SARS-CoV-2 infection. Unfortunately, SARS-CoV-2 patients show very similar imaging features to patients with other pulmonary infections. Thus, it is very difficult to develop AI software that is able to differentiate SARS-CoV-2 from other viral pneumonias, mainly as regards influenza viruses.

Starting from these considerations and given the increasing role of radiologists in the diagnosis and management of SARS-CoV-2 patients, further considerations must be performed about the use of AI and, in particular, deep learning, in the fight against this virus. In this context, researchers have developed deep learning algorithms that are able to quickly identify patterns associated with SARS-CoV-2 infection [70]. Moreover, an ACR Data Science Institute use case for AI algorithms that addresses the pertinent characteristics of SARS-CoV-2 is available. These use cases are created by radiologists and are meant to be used as open-source guidelines for data scientists who wish to design neural network architecture to identify features or patterns of radiographic pathology [71].

Currently, the most promising study about the use of AI in the early diagnosis of SARS-CoV-2-related diseases was published by Xueyan Mei et al. in Nature Communication [72]. In this paper, the authors used AI algorithms to integrate chest CT findings with clinical symptoms, exposure history, and laboratory testing to perform a very early diagnosis of SARS-CoV-2 infection in positive patients [72]. In a test set of 279 patients, the AI system proposed by Mei and colleagues showed equal sensitivity as compared to a senior thoracic radiologist [72]. The use of the AI system also improved the finding of SARS-CoV-2 positive patients who presented with normal CT scans, correctly identifying 17 of 25 (68%) patients. Surprisingly, a team of expert radiologists diagnosed all these patients as SARS-CoV-2 negative. Lastly, in patients with CT scans and associated clinical history, the AI system was useful in the early detection of SARS-CoV-2 infection [72].

As regards non-pulmonary pathologies, myocarditis is the main pathology related to SARS-CoV-2 and also represents the most frequent cause of death in patients with SARS-CoV-2.

Myocarditis is defined as an inflammatory disease of the heart that causes myocardial injury without an ischemic cause [73]. Despite only a few cases of all viral myocarditis being related to human virus infections, some coronavirus has been associated to myocarditis in patients of all age groups [74,75]. Interestingly, viral RNAs of both MERS-CoV and SARS-CoV, which are close relatives of SARS-CoV-2, have been identified in the heart animal models, suggesting that frequently, coronaviruses possess cardiotropism [76,77].

Despite several studies highlighted hearth impairment, the incidence of hearth involvement in SARS-CoV-2 patients is unclear. However, up to 7% of the SARS-CoV-2-related deaths are attributable to myocarditis [78].

In this regard, the American Heart Association (AHA) recently recommended more extensive tests for patients with early signs of myocarditis, mainly cardiac imaging analysis such as echocardiogram or Cardiovascular Magnetic Resonance (CMR) [79]. Although CMR is considered more accurate and sensitive than echocardiography, its use in this pandemic can be limited due to its restricted out-of-hours availability [78]. Myocardial edema and/or scarring were the most frequent CMR signs observed in SARS-CoV-2 patients affected by myocarditis [31,80,81]. If CMR analysis cannot be performed, both cardiac CT scan with contrast enhancement and ECG gating may represent an effective alternative. In addition, it is reported that CT or MRI of the head can be useful to detect strokes in SARS-CoV-2 patients that show acute neurological symptoms [82,83]. However, the management of SARS-CoV-2 patients showing brain involvement is even more challenging. In fact, neuroimaging by CT cannot always provide a diagnostic certainty [84]. Conversely, lumbar punctures could provide important data, but this diagnostic test is rarely positive [84]. Clinical imaging of the brain in patients affected by SARS-CoV-2 commonly occurs in the last phase of the infection with very different phenotypes (MRI, CTh, demyelinating lesions and encephalopathy) [85]. Nevertheless, the presence of SARS-CoV-2 in the cerebrospinal fluid or its load can be significantly reduced in the late phases of the infection, thus resulting in it not being detectable with conventional analyses. Since brain involvement in these patients can be severe and fatal, prompt treatment could be required. Recently, demyelinating lesions detected by MRI head scan were described by Zanin et al. [86] in four SARS-CoV-2 patients, demonstrating the possible role of neuroimaging in the management of positive SARS-CoV-2 patients both symptomatic or not.

## 11. The Role of Nuclear Medicine in SARS-CoV-2-Related Diseases

^18^Fluorine fluorodeoxyglucose positron emission tomography/CT (^18^F-FDG PET/CT) has been proposed as a non-invasive imaging method for detecting infectious or inflammatory diseases [87,88]. The ability of ^18^F-FDG PET/CT to identify sites of inflammation and infection is mainly related to the glycolytic activity of cells involved in the inflammatory response [89]. It has been demonstrated that cells involved in infection and inflammation, especially neutrophils and the monocyte/macrophage family, are able to express high levels of glucose transporters and hexokinase activity. Notably, ^18^F-FDG PET/CT may detect early pathophysiological changes in affected tissues in patients with infectious or inflammatory diseases, and these functional changes may occur before anatomical changes are detected by conventional imaging techniques [89]. In the literature, sufficient evidence-based data on the utility of ^18^F-FDG PET/CT in the diagnosis and management of several infectious and inflammatory diseases already exist [90].

Therefore, ^18^F-FDG-PET/CT plays an important role in assessing infectious and inflammatory lung diseases, detecting involved lung segments, estimating the extent of lung involvement, monitoring progression and treatment responses, and following up [91].

Recently, some case reports and small case series have shown ^18^F-FDG PET/CT findings in patients with acute respiratory disease caused by SARS-CoV-2 infection (see Table 5). In the largest series described by Qin et al. [92], ^18^F-FDG PET/CT results from four patients with suspected severe acute respiratory syndrome coronavirus-2 (SARS-CoV-2) infection were described. The patients were admitted to hospital with respiratory symptoms and fever when the SARS-CoV-2 outbreak was still unrecognized and the virus infectivity was unknown. All the patients had typical chest CT imaging features of SARS-CoV-2 pneumonia. In the ^18^F-FDG PET/CT scan, lung lesions were characterized by increased ^18^F-FDG uptake, and there was evidence of lymph node involvement [92]. Conversely, disseminated disease was absent, suggesting that SARS-CoV-2 has pulmonary tropism. In their conclusion, the authors suggested a potential clinical usefulness for ^18^F-FDG PET/CT in patients with suspected SARS-CoV-2 infection, especially at the early stages, when clinical symptoms are not specific and differential diagnosis is challenging.

In a paper by Setti et al. [93], the authors retrospectively reviewed the cases of patients who showed pulmonary involvement unrelated to cancer metastases. Among the 13 scans, five cases with imaging findings suspicious for viral infection were detected. Peripheral lung consolidations and/or ground-glass opacities in two or more lobes were found. Lung abnormalities displayed increased ^18^F-FDG uptake (maximum standardized uptake value [SUVmax] 4.3–11.3) [93]. All the patients on the day of PET/CT acquisition were asymptomatic, and they did not have fever or cough. In view of the PET/CT findings, home isolation, symptom surveillance, and treatment (in 3/5 patients) were indicated. At 1-week follow-up, two out of five patients experienced the onset of mild respiratory symptoms. The ^18^F-FDG PET/CT result revealed the presence of bilateral, diffuse, and intense FDG uptake in the lower lobes (right lower lobe SUVmax = 5.9; left lower lobe SUVmax = 7.9; SUVmean of the liver = 2.0) and less intense uptake in the remaining lobes. The FDG uptake corresponded to the peripherally predominant ground-glass opacities observed in low-dose computed tomography (CT) without contrast media administration. The ^18^FDG-PET/CT scan can identify probable SARS-CoV-2 disease in the absence or before symptom onset and can guide patient management [93].

Moreover, in a case report by Polverari and colleagues [70], the authors declared that during a routine CT scan performed in February 2020, a centimetric nodule in the left superior lobe that was suspected of being malignant was found in a 73-year-old male patient who underwent medium lobe resection for pT2aN0 non-small cell lung cancer in April 2016, without administration of adjuvant therapies [94]. Functional imaging with ^18^F-FDG PET/CT was requested by the tumor board to evaluate the nodule metabolism. ^18^F-FDG PET/CT was scheduled on 18 March 2020, 27 days after the outbreak of SARS-CoV-2 in Italy. During the triage procedures required for the prevention of SARS-CoV-2 infection, the patient’s body temperature was less than 37.5 °C, and he presented neither cough nor wheezing nor shortness of breath. The patient declared no exposure to suspected infected people, and he was a non-smoker with no cardiovascular comorbidities. However, the ^18^F-FDG PET/CT result revealed the presence of bilateral, diffuse, and intense FDG uptake in the lower lobes (right lower lobe SUVmax = 5.9; left lower lobe SUVmax = 7.9; SUVmean of the liver = 2.0) and less intense uptake in the remaining lobes [94]. The FDG uptake corresponded to peripherally predominant ground-glass opacities observed in low-dose CT without contrast media administration. An increased uptake of ^18^F-FDG in the mediastinal lymph nodes was also observed (SUVmax = 5.6 in the right lower paratracheal node). No pleural effusion was noted. The solitary nodule in the left superior lobe did not reveal relevant ^18^F-FDG uptake. The authors interpreted the PET scan results as active inflammatory processes, with a CT pattern highly suggestive of ongoing SARS-CoV-2 pneumonia. The patient was tested with a RT-PCR reaction that revealed a positive result, and he was subsequently isolated. Similarly, the studies of both Colandrea et al. and Habouzit and colleagues identified asymptomatic SARS-CoV-2 patients before an RT-PCR test by ^18^F-FDG PET/CT analysis [95,96].

Zou et al. recently reported an ^18^F-FDG-PET/CT case of a PCR-confirmed SARS-CoV-2 patient [97]. FDG uptake was observed in ground-glass opacities with areas of focal consolidation in the right lung (SUVmax = 4.9), in the right paratracheal, and right hilar lymph nodes. Notably, indications for bone marrow involvement were seen. Czernin et al. recently published a ^18^F-FDG-PET/CT scan of a 53-year-old patient with a neuroendocrine tumor of the pancreas, who was referred for restaging [98]. At the time the PET scan was performed, the patient was completely asymptomatic. On the PET scan, a new hypermetabolic region in the right upper and lower lobe (SUVmax = 5.5) was observed, which was in topographic correlation to predominantly peripheral and subpleurally located ground-glass opacities with incipient, partly round-shaped consolidations. The findings were attributed to an atypical inflammation; later, SARS-CoV-2 infection was confirmed [98].

Another potential application of ^18^F-FDG-PET/CT in SARS-CoV-2 could be to monitor treatment response and help predict recovery time. The data provided by Qin et al. suggest a trend that higher FDG uptake in SARS-CoV-2-induced pulmonary lesions may be correlated with longer healing times, as one patient with an SUVmax of 4.6 recovered approximately 17 days after the onset of symptoms, while another patient with a SUVmax of 12.2 recovered more than 26 days after the appearance of the first symptoms [92]. The patient described by Zou et al. had a SUVmax of 4.9 in a pulmonary lesion and recovered 15 days after the first symptoms occurred [97]. Certainly, these are just case observations, which need to be properly characterized in larger patient cohorts before conclusions can be drawn. To evaluate the potential predictive capability of PET for outcome, quantitative parameters could be correlated with time on ventilation or death.

Although diagnostic reports of nuclear medicine performed in SARS-CoV-2 patients are sparse, the first available case report indicate the usefulness of ^18^F-FDG-PET/CT to visualize inflamed or infected lung areas in SARS-CoV-2. Das and colleagues reported on a patient with MERS-CoV infection, who developed pneumonia with a severe radiographic deterioration pattern, multiple FDG-avid areas on ^18^F-FDG-PET/CT, corresponding with nodules and cavities [98,99]. Chefer et al. visualized the immune response to MERS-CoV 5 days after viral challenge with ^18^F-FDG-PET/CT in a non-human primate model, showing FDG-avid mediastinal and axillary lymph nodes [100]. Interestingly, no changes in the CT image, body temperature, body weight, and blood glucose concentrations were observed after viral exposure. However, FDG uptake in lymph nodes at Day 5 after viral exposure was accompanied by a slight increase (within the normal range) in circulating monocytes. As monocytes play an important role in the immune response to viral infections, the reported correlation between FDG uptake in lung-draining lymph nodes and monocyte count is not surprising. In that respect, it would be interesting to assess the composition of the pulmonary manifestations of SARS-CoV-2 with immune PET imaging, in order to characterize the involved immune cell subsets [101]. Muehe et al. recently performed PET imaging using ^89^Zr-labebed Feraheme, an FDA-approved iron oxide nanoparticle, in non-human primates to visualize resident macrophages and monocyte trafficking [102]. They reported that areas of acute inflammation and their draining lymph nodes could be visualized clearly up to 14 days post injection.

In line with the findings presented by Chefer et al. [103], Wallace and colleagues performed ^18^F-FDG-PET/CT imaging of activated lymphoid tissues during simian-human immunodeficiency virus infection in rhesus macaques, and they reported that FDG uptake in lymph nodes can precede fulminant viral replication. The authors also concluded that ^18^F-FDG-PET can detect even subtle changes in host immune response to contain a subclinical MERS-CoV infection [103]. For SARS-CoV-2 management, these observations suggest that ^18^F-FDG-PET/CT imaging might play a role in the early stages of the disease, when clinical symptoms are unspecific and differential diagnosis is challenging. With an increasing number of infected people, nuclear medicine physicians may also be confronted with PET and CT signs of SARS-CoV-2 as incidental findings in patients referred for other clinical questions, especially when patients are completely asymptomatic or in cases reported by Czernin et al. [98]. Therefore, it is important to be alert and report these signs to the referring physicians.

Although there is no definitive evidence, asymptomatic patients who present typical radiologic CT patterns and positive FDG uptake should be promptly tested and strictly monitored, because a sudden worsening of clinical conditions is possible.

Remarkably, ^18^F-FDG-PET/CT imaging of SARS-CoV-2 patients can be used for evaluating FDG-uptake pattern in non-lung sites elsewhere in the body. Indeed, it is known that the infection by SARS-CoV-2 can provide damages to other organs such as heart, the gastrointestinal tract, kidneys, and/or bone marrow [104,105,106]. Due to the absence of solid data about the non-lung localization of SARS-CoV-2, some hypotheses have been derived by previously studies on other coronaviruses, such as MERS-CoV and SARS-CoV infection. Chefer and colleagues described an increase of the uptake of FDG in the bone marrow in MERS-CoV positive patients [102]. Similarly, Zou et al. reported bone marrow involvement and FDG uptake in a SARS-CoV-2 patient [97]. In addition, Galougahi et al. [107], for the first time, showed hypometabolism of the left orbito-frontal cortex by FDG-PET/CT scan in a SARS-CoV-2 patient affected by anosmia. Specifically, the authors observed a reduction of the Standardized Uptake Value (SUV) of the left side (9.5) with respect to the SUV of the right side (10.0). In this context, it is important to remember that anosmia is one of the most frequent non-pulmonary symptoms in SARS-CoV-2 positive patients [108]. Starting from these preliminary results, is it possible to hypothesize the use of PET/CT scan as a whole-body non-invasive readout to assess chronic and concomitant organ damage in SARS-CoV-2 positive patients. For what concerns heart and brain SARS-CoV-2 related damage, ^18^F-GE180 PET analysis can be used for the concomitant detection of heart–brain inflammation by targeting the mitochondrial translocator protein [109].

In addition, nuclear medicine showed the potential to provide new molecular data about the use of nonsteroidal anti-inflammatory drugs in SARS-CoV-2 patients [110] by directly depicting cyclooxygenase-2 (COX2)-involvement, using established COX2 inhibitory radiopharmaceuticals [111]. The use of radiolabeled drugs, to investigate molecular mechanisms involved in the SARS-CoV-2 infection, could target the cytokine signaling pathway involved in the cellular internalization of SARS-CoV-2, such as chemokine receptor CXCR4, interleukin IL-6, and fibroblast activation protein inhibitors, to address post-inflammatory fibrosis, or inhibitors of the type 1 angiotensin-II receptor ATR1 [112]. It is noteworthy that the development of novel radiopharmaceuticals may also be directed against the angiotensin-converting-enzyme-2 (ACE2), which is the entry receptor for SARS-CoV-2. Radiolabeled ACE2-receptor antagonist has already been developed for autoradiography analysis [113], laying the foundation for PET tracer development that can provide essential information in the study of SARS-CoV-2 infection.

However, the identification of new biomarkers for the development of new radiolabeled drugs for both the diagnosis and therapy of SARS-CoV-2 requires an intense collaboration with the pathology departments. Indeed, it is important to identify the in situ expression (on bioptic specimens) of biomarkers related to the SARS-CoV-2 activities, inflammatory response, and/or tissue damage, before developing potential radiolabeled drugs that are useful in the fight against the SARS-CoV-2 pandemic.

## 12. Conclusions

Currently, the SARS-CoV-2 pandemic represents the focus of the biomedical research worldwide. The identification of the molecular events related to the SARS-CoV-2 infection, as well as the characterization of its clinical features, could put an end to this dramatic health emergency. In this scenario, the interpretation of histopathological data in light of the clinical imaging characteristics of SARS-CoV-2 infection can provide the scientific rationale to develop diagnostic and therapeutic protocols that are capable of improving the management of infected patients. Specifically, morphological and molecular analysis of SARS-CoV-2 infected tissues could highlight new useful prognostic and predictive biomarkers for in vivo investigations.

## Figures and Tables

**Figure 1 ijms-21-06960-f001:**
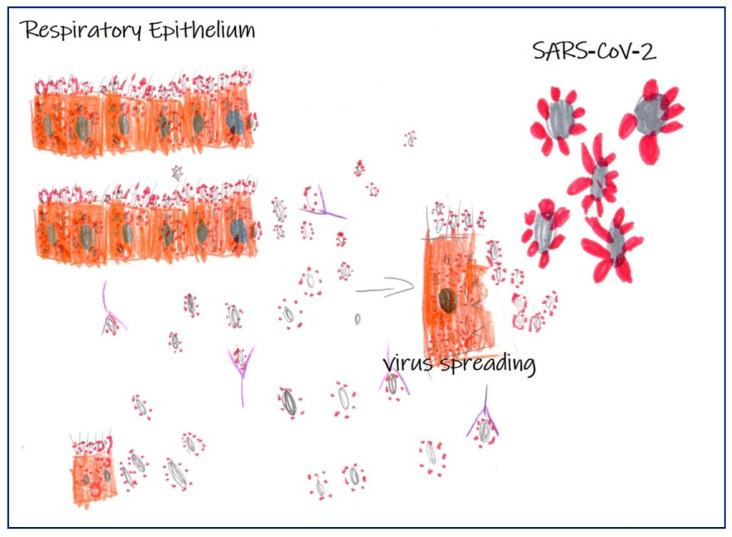
Scheme of severe acute respiratory syndrome coronavirus-2 (SARS-CoV-2) infection in respiratory epithelium. Image shows infection of SARS-CoV-2 in respiratory epithelium, virus spreading, and the antibody response (Martina Gioia Simeca, 5-year-old).

**Table 1 ijms-21-06960-t001:** Histopathological characteristics of lungs infected by SARS-CoV-2.

Characteristics	SARS-CoV-2 Infection
Gross Pathology	(a) multiple areas of congestion (b) edematous lungs (c) >gross weights (d) pulmonary embolism
Microscopic examination	(a) diffuse alveolar damage (b) severe capillary congestion (c) interstitial mononuclear cell infiltrates (d) multinucleated syncytial cells with atypical enlarged pneumocytes (e) microthrombosisa
Pathogenesis	Combination of direct virus-induced cytopathic effects, immunologic injury, and microvascular damage induced by cytokines

**Table 2 ijms-21-06960-t002:** Most important histological lesions observed during autopsy of patients who died from SARS-CoV-2 infection.

**Lung**	**References**
Diffuse alveolar damage (histological hallmark of SARS-CoV-2 infection)	[15,16,17,18,19,20,21,22,23,24]
Focal vasculitis and capillaritis associated to microthrombosis as direct viral effect	
Thrombosis of large and medium-size pulmonary, related to SARS-COV-2-associated coagulopathy (likely secondary to an endothelial damage related to direct viral infection of the endothelial cells) or deriving from the deep veins of the lower extremities. Superimposed bronchopneumonia as result of bacterial superinfection	
**Heart**	
Myocardial damage and myocarditis associated with increase in troponin levels, related to (a) direct myocardial infection by SARS-CoV-2 (b) hypoxemia due to respiratory failure and (c) inflammatory response correlated to the severe systemic inflammation status. Acute vasculitis of the intramyocardial vessels	[25,26,27,28,29,30]
**Kidney**	
Acute tubular injury involving mainly the proximal tubules, probably related to direct infection of kidney by SARS-CoV-2	[30,32]
**Skin**	
Urticarial rashes and papulovesicular exanthems (cause not yet known)	[33,34,35,36,37,38,39,40]
Livedoid purple lesions and acrocyanosis	
Kawasaki disease	
**Central Nervous System**	
Aspecific acute hypoxic damage in the brain and cerebellum (molecular test in sections of brain tissue were positive for the virus, but not immunohistochemistry)	[41]
**Liver**	
Sinusoidal dilatation with lymphocytic infiltration and steatosis (cause not yet known)	[20]
**Adrenal**	
Acute fibrinoid necrosis of arterioles (cause not yet known)	[42]
**Testis**	
Seminiferous tubular injury, mild lymphocytic inflammation (cause not yet known)	[43]

**Table 3 ijms-21-06960-t003:** Table reported the main studies with the computerized tomography (CT) characterization of SARS-CoV-2 patients.

	Patients	Sex	Age (mean)	Type of Study	References
Bernheim A et al.	121	61 M; 60 W	45 ± 16	R	[45]
Pan F et al.	21	6 M; 15 W	40 ± 9	R	[46]
Shi H et al.	81	42 M; 39 W	49.5 ± 11	R	[47]
Fang Y et al.	51	29 M; 22 W	45	R	[48]
Yoon SH et al.	9	4 M; 5 W	54	R	[50]
Li Y et al.	53	29 M; 24 W	58 ± 17	R	[51]
Wei J et al.	1	1 W	40	CR	[52]
Hu Z et al.	24	/	/	R	[53]
Chen Z et al.	98	M 52; W 46	43 ± 17.2	R	[54]
Chen N et al.	99	M 67; W 32	55.5 ± 13.1	R	[31]
Huang C et al.	41	M 30; W 11	49	R	[11]
Wang D et al.	138	M 75; 63 W	/	R	[12]
Chung M et al.	21	M 13: W 8	51 ± 14	R	[56]
Song F et al.	51	M 25; W 26	49 ± 16	R	[58]
Ai T et al.	1014	M 467; W 547	51 ± 15	R	[59]
Ng MY et al.	18	M 13; W 8	56	R	[61]

R: retrospective CR: case report.

**Table 4 ijms-21-06960-t004:** CT imaging features of patients affected by SARS-CoV-2.

CT Findings	Number of Studies	Number of Patients (%)
*Patterns of the lesion*		
Ground-glass opacity with consolidation	60	768 (18%)
Ground-glass opacity	60	2482 (65%)
Consolidation	60	1259 (22%)
Crazy paving pattern	24	575 (12%)
Reversed halo sign	24	146 (1%)
*Other signs in the lesion*		
Interlobular septal thickening	23	691 (27%)
Air bronchogram sign	23	531 (18%)
*Distribution*		
Bilateral	48	3952 (80%)
Unilateral	48	641 (20%)
Right lung	8	48 (62%)
Left lung	8	29 (38%)
*Number of lobes involved*		
One lobe	13	278 (14%)
Two lobes	13	299 (11%)
Three lobes	13	250 (13%)
Four lobes	13	212 (15%)
Five lobes	14	384 (34%)
More than one lobe	14	1145 (76%)
*Lobe of lesion distribution*		
Left upper lobe	14	731 (74%)
Left lower lobe	20	504 (46%)
Right upper lobe	19	455 (40%)
Right middle lobe	15	326 (38%)
Right lower lobe	17	784 (74%)
*Other findings*		
Pleural effusion	60	94 (1.6%)
Lymphadenopathy	60	21 (0.7%)
Pulmonary nodules	22	262 (9%)

Note—data are from reference [44].

**Table 5 ijms-21-06960-t005:** Table reported the main nuclear medicine studies about SARS-CoV-2 pandemic.

	SARS-CoV-2 Positive	Imaging Analysis	Type of Study	Reference
Qin et al.	4	^18^F-FDG PET/CT	R	[92]
Setti et al.	13	^18^F-FDG PET/CT	P	[93]
Polverari et al.	1	^18^F-FDG PET/CT	CR	[94]
Colandrea et al.	5	^18^F-FDG PET/CT	CS	[95]
Habouzit et al.	1	^18^F-FDG PET/CT	CR	[96]
Zou et al.	1	^18^F-FDG PET/CT	CR	[97]

R: retrospective P: prospective CR: case report CS: case series.

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
