# Peer review of "Imaging Diagnostics and Pathology in SARS-CoV-2-Related Diseases"

_ijms, 2020, doi:10.3390/ijms21186960_

Round 1

Reviewer 1 Report

It might be the right time to review on imaging diagnostics and pathology of COVID-19.

However, the manuscript is lack of details.

There should be summarized and organized analysis of previously published works.

In the case of table 1, the author just listed the organs and their descriptions.

This is one of the previously published studies. (I’m not related with the authors)

https://www.ncbi.nlm.nih.gov/pmc/articles/PMC7156158/

The various imaging signs of COVID-19 were well summarized as a table.

The author should summarize such data as organized tables.

It’ll be better with tables of CT, X-ray, and other imaging methods.

Currently, the tables in this manuscript are not so informative.

Author Response

Before we begin the point by point review of the list of concerns, we would like to thank the Reviewer for their comments on how to improve the manuscript, which has been revised accordingly, as well as the Editors for calling for a new submission of an improved version of our manuscript.

Reply to Reviewer 1

It might be the right time to review on imaging diagnostics and pathology of COVID-19.

Reply: we would like to thank the Reviewer for expressing interest in our work, and for their availability to review a revised version of our manuscript.

There should be summarized and organized analysis of previously published works.

In the case of table 1, the author just listed the organs and their descriptions.

This is one of the previously published studies. (I’m not related with the authors)

https://www.ncbi.nlm.nih.gov/pmc/articles/PMC7156158/

The various imaging signs of COVID-19 were well summarized as a table.

The author should summarize such data as organized tables.

It’ll be better with tables of CT, X-ray, and other imaging methods.

Currently, the tables in this manuscript are not so informative.

Reply: Thank you for this pointing out.  In the new version of our manuscript we added two tables reporting information about the imaging diagnostic data. In addition, we improved the ex-table 1 (now table 2) and added a further table with histopathological characteristics of SARS-CoV-2 Infection in the lungs.

Reviewer 2 Report

The article submitted by Scimeca and co-workers is a well written review on SARS-CoV-2 pathology. The SARS-CoV-2 subject is very new, therefore any additional information and review on the current state of the knowledge may me crucial for a further research.

The target audience of the article are medicinal doctors and biologists. If MD, should understand the article, the probability of the observation of the histological observables should me more pointed. The good place to do so is Table 1.

The reviewer is also missing the references in Table 1 (copy form text).

The age of the patients is also crucial, and according to the reviewer, those data are treated in an insufficient way.

General issue for the Table 2, is the same as in Table 1. What is the probability of the successful diagnosis of the infection by using nuclear medicine.

The figure painted by Martina is beautiful, therefore the reviewer has a hope that she will become a scientist not an artist.

Author Response

Before we begin the point by point review of the list of concerns, we would like to thank the Reviewer for their comments on how to improve the manuscript, which has been revised accordingly, as well as the Editors for calling for a new submission of an improved version of our manuscript.

Reply to Reviewer 2

The article submitted by Scimeca and co-workers is a well written review on SARS-CoV-2 pathology. The SARS-CoV-2 subject is very new, therefore any additional information and review on the current state of the knowledge may me crucial for a further research.

Reply: we would like to thank the Reviewer for expressing interest in our work, and for their availability to review a revised version of our manuscript.

The target audience of the article are medicinal doctors and biologists. If MD, should understand the article, the probability of the observation of the histological observables should me more pointed. The good place to do so is Table 1. The reviewer is also missing the references in Table 1 (copy form text).

Reply: Thank you for this pointing out.  In the revised for of our manuscript we improve the information reported in the table 1 (now table 2) also adding the relative references. In addition, we added a further table with the histopathological characteristics of lung infected by SARS-CoV-2.

The age of the patients is also crucial, and according to the reviewer, those data are treated in an insufficient way.

Reply: Thank you for this suggestion.  We added a further table with age, gender and type of study of patients investigated by CT analysis.

General issue for the Table 2, is the same as in Table 1. What is the probability of the successful diagnosis of the infection by using nuclear medicine.

Reply: Thank you for this pointing out.  In the new version of the manuscript we added Two tables for imaging diagnostic data. Specifically, we added a table with the main characteristics of CT investigations. As concern the nuclear medicine, data are limited yet to reported solid data about the probability of the successful diagnosis of the infection.

The figure painted by Martina is beautiful, therefore the reviewer has a hope that she will become a scientist not an artist.

Reply: Thank you for this comment.  We agree with the reviewer about Martina Gioia. Currently, the dream of Martina Gioia is to become a scientist.

Round 2

Reviewer 1 Report

Much has improved from the previous one.

There are some minor comments

Minor comments

Page 8, Line 336-337

Died for -> deceased by

Table 3

The comma should be replaced with a period (ex 55,5 ->55.5)

Table 4

“Reference 44” may not proper citation. Check the format.

Table 5

In the “Patients” column, Qin et al. may not name of patients. It should be the authors of the reference. The column should be removed or replaced with an informative one.